# Fecal Microbiota Transplantation as a Treatment for Granulomatous Colitis in a French Bulldog: A Case Report

**DOI:** 10.3390/microorganisms13020366

**Published:** 2025-02-08

**Authors:** Felipe S. Romano, Maria A. Lallo, Raquel S. Romano, Letícia P. S. Isidoro, Mariane R. Cardoso, Lylian C. Sodré, Alessandra Melchert, Priscylla T. C. Guimarães-Okamoto, Maria C. F. Pappalardo, Andressa R. Amaral, Thiago H. A. Vendramini

**Affiliations:** 1Department of Experimental and Ambiental Pathology and Research, Paulista University, São Paulo 04026-002, SP, Brazil; felipe.med.vet@hotmail.com (F.S.R.); maria.lallo@docente.unip.br (M.A.L.); raquelromano.vet@hotmail.com (R.S.R.); 2Ferogastro—Clinic of Gastroenterology in Small Animals, São Paulo 04062-003, SP, Brazil; leticiagastrovet@gmail.com (L.P.S.I.); mariruizcardoso@gmail.com (M.R.C.); lyliancristina.cs@gmail.com (L.C.S.); 3Department of Veterinary Small Animal Internal Medicine, School of Veterinary Medicine and Animal Science, São Paulo State University, Botucatu 18618-687, SP, Brazil; alessandra.melchert@unesp.br (A.M.); tatiana.okamoto@unesp.br (P.T.C.G.-O.); 4Pet Nutrology Research Center, Department of Animal Nutrition and Animal Production, School of Veterinary Medicine and Animal Science, University of São Paulo, Pirassununga 13635-900, SP, Brazil; carolpappalardo@usp.br (M.C.F.P.); andressa.r.amaral@gmail.com (A.R.A.)

**Keywords:** chronic enteropathy, dog, dysbiosis index, histiocytic ulcerative colitis, microbiome

## Abstract

Granulomatous colitis, or ulcerative colitis, is an infectious and inflammatory disease that primarily affects the colon and occasionally extends to the ileum, particularly in young Boxer and French Bulldogs. Unlike typical chronic colitis in small animals, the early onset of the disease suggests a genetic predisposition. The condition is characterized by the overgrowth of *Escherichia coli*, specifically the enteroinvasive variant, which displaces beneficial gut bacteria, contributing to its infectious nature. Secondary dysbiosis and chronic-active inflammation involving histiocytes and other leukocytic infiltrates are prominent features. Clinical manifestations include chronic diarrhea with blood and mucus, frequent tenesmus, and pain, with variable degrees of weight loss depending on disease severity. The final diagnosis is based on clinical history (chronic diarrhea with hematochezia), macroscopic findings from colonoscopy (edema, ulcers, and wall hyperplasia), histopathology (presence of histiocytes), and *Escherichia coli* growth in culture from a colon fragment. Treatment is guided by colon antibiograms, which often require prolonged antibiotic therapy. Fecal microbiota transplantation (FMT) has emerged as a potential treatment, either as a primary intervention or adjunctive therapy, for conditions such as acute enteritis (e.g., canine parvovirus), dysbiosis, and chronic enteropathies. However, its application to modulate the microbiota and reduce inflammation in granulomatous colitis, potentially leading to longer intervals between relapses, remains an area of ongoing investigation. This is a case report of a French Bulldog diagnosed with ulcerative colitis accompanied by dysbiosis and refractory to standard treatments but sensitive and partially responsive to amikacin. The patient achieved control and sustained improvement in fecal scoring following fecal transplantation. This approach prevented the need for additional antibiotic therapy, ensuring clinical amelioration alongside microbiome restoration.

## 1. Introduction

First described in the United States in 1965 in humans, ulcerative colitis remains as an emerging disease due to evolving understandings of its pathology and increasing reports in dogs [1]. Its presentation can mimic other conditions due to nonspecific manifestations, potentially leading to delayed diagnosis and increased chronicity. Initially more prevalent in Boxers, ulcerative colitis is now increasingly diagnosed in French Bulldogs, likely due to their rising popularity and potential breed predisposition. Screening is essential for any young animal presenting with a history of chronic diarrhea to rule out causes such as parasitism, viral and bacterial enteritis, food hypersensitivity, exocrine pancreatic insufficiency, and gastrointestinal foreign bodies [2,3,4].

Other dog breeds described include the Alaskan Malamute, English Mastiff, American Staffordshire Terrier, Dachshund, and English Bulldog [5]. More recently, it was reported in a cat [6]. Genetic predisposition is plausible but not fully understood [3,5].

Ulcerative colitis is believed to have a bacterial origin, caused by *Escherichia coli* infection. While these bacteria are typically insignificant under conditions of eubiosis, in dogs with ulcerative colitis, genetic variations in this disease enhance its virulence and pathogenic potential, endowing it with invasive potential and immune evasion mechanisms. These factors may justify the chronic inflammatory-infectious state and resistance to antimicrobial treatments [3,7,8].

Affected animals typically present with diarrhea accompanied by blood and mucus, often showing significant tenesmus. Bloating and flatulence may also occur. Additionally, symptoms such as vomiting, weight loss, and lethargy have occasionally been observed. Appetite variability is common [3,7,8,9,10].

Serum findings in affected animals can vary or even be absent. Anemia, neutrophilia, elevated urea, monocytosis, hypocobalaminemia, and other changes may occur, reflecting blood loss, dysbiosis, or inflammation. Typical findings of intestinal malabsorption, common in small intestinal diseases, such as hypoalbuminemia and hypocalcemia, may or may not be present [8,10,11].

Ultrasonography can evaluate motility, intestinal wall thickness, and the presence of gas, although its application in colon assessment is limited. Abdominal distension and other severity-dependent clinical signs, such as pain or discomfort, can also be observed through physical examination [12]. This examination assists in ruling out foreign bodies and other undesirable conditions such as hepatic, renal, and pancreatic diseases. Although uncommon, the use of contrast agents can enhance diagnostic accuracy [3,13].

Colonoscopy is the preferred diagnostic procedure, allowing for intraluminal evaluation to observe macroscopic findings, such as common structural damage in ulcerative colitis, and the collection of mucosal samples from the colon, and if feasible, the ileum for histopathological examination and culture with antibiograms [3,8]. Histological staining techniques employed include hematoxylin–eosin (H&E) and periodic acid-Schiff (PAS), which assess the extent of lymphocytic and histiocytic infiltrates along with macrophage staining, ensuring differentiation from less common enteritis causes such as fungi or *Leishmania* sp. [10,12,14]. Immunohistochemistry has been described as a diagnostic method in specific cases [15].

Antibiotics constitute the primary treatment for the disease, typically administered for 30 to 90 days based on the colon culture results, with careful monitoring for side effects [9,12]. Currently, there is not sufficient scientific evidence regarding the therapeutic or palliative role of probiotics for this condition [7,16,17].

Anti-inflammatories are not always essential but can help alleviate signs and symptoms. Mesalazine, a salicylate (non-steroidal anti-inflammatory drug), is widely used in dogs with chronic colitis of non-infectious origin. Glucocorticoids like prednisolone and immunosuppressants such as cyclosporine are not typically required in most cases [18], but may be beneficial in specific cases, particularly when inflammatory bowel disease coexists [2,4,7].

## 2. Case Report

A 1.5-year-old male French Bulldog was presented body condition score (3 out of 9), chronic and severe hematochezia, tenesmus, and flatulence. After a diagnostic screening that did not identify a cause and considering the absence of response to a hydrolyzed diet, the dog underwent a colonoscopy for further evaluation, which showed wall edema, bleeding, relative lumen stenosis, and hyperemia (Figure 1).

Aerobic culture of the colon revealed the presence of *Escherichia coli*, and histological staining confirmed granulomatous colitis through positive macrophage markers and the presence of histiocytes. Treatment with mesalazine was ineffective. Adding oat fiber to the diet resulted in a slight reduction in tenesmus, while probiotics (*Lactobacillus* sp. and *Enterococcus* sp.) primarily reduced flatulence.

Based on the antibiogram, the only therapeutic option identified was amikacin (Fresenius Hemocare Brasil Ltda., Sao Paulo, Brazil), an aminoglycoside antibiotic. To date, only one study has reported the use of this medication in dogs with colitis [19]. The dog received amikacin at a dose of 10 mg/kg/day for 15 days, with renal function monitored throughout the treatment. Clinical improvement was observed within three days of initiating therapy, with the dog passing normal feces for the first time. However, symptoms recurred three days after completing the treatment.

The patient’s dysbiosis index was assessed via quantitative PCR (qPCR) according to the protocol previously described in [20]. After the DNA from a fecal sample was extracted, a quantitative polymerase chain reaction was performed using primers for *Faecalibacterium, Fusobacteria, Blautia, Turicibacter, Escherichia coli, Clostridium hiranonis*, and *Streptococcus*. Results are expressed as the log amount of DNA for each bacterial group per 10 ng of total isolated DNA. The dysbiosis index is a single numerical value quantifying the imbalance in microbial communities, calculated as the difference in Euclidean distances between the test sample and the centroids of healthy and diseased classes, indicating its proximity to either group. Values below 0 are considered normal, 0–2 mildly increased, and above 2 significantly increased. The patient’s results yielded a final score of +4.1 (Table 1). A second 15-day course of amikacin was administered; however, symptoms worsened one day after its completion.

Subsequently, fecal microbiota transplantation (FMT) was performed. The dog was fasted for six hours before and four hours after the procedure to prevent premature bowel evacuation. The FMT was administered via rectal enema using a urethral catheter (size 14, 25 cm in length) (Figure 2) at a dose of 5 g of frozen donor stool per kg of body weight according to [21].

The donor dog was a 4-year-old spayed female Shih Tzu, with a body weight of 5 kg, clinically healthy. The donor was clinically healthy, free of intestinal parasites, bacterial pathogens, and extended-spectrum beta-lactamase-resistant *Escherichia coli*. Additionally, the donor exhibited a high expression of *Clostridium hiranonis* (Table 2).

Feces normalization occurred within 48 h of the first FMT (Figure 3), and a second transplant was performed two weeks later. The dog gained weight and has remained asymptomatic for nearly three months. A follow-up dysbiosis index score of −2 (indicative of eubiosis) was obtained, closely resembling the donor’s microbiota including the increased *Clostridium hiranonis* population (Table 1).

## 3. Discussion

The breed and age of the patient in this case report aligned with the epidemiological data on granulomatous colitis reported in the literature. [3,4]. The use of serum and fecal screening to rule out parasitic diseases aligned with the recommendations by Romano and Lallo [22], effectively excluding potential differential diagnoses for chronic diarrhea such as giardiasis and trichuriasis [23]. Colonoscopy remains the preferred diagnostic method for its minimally invasive nature and ability to provide intraluminal inspection [13].

Antibiotics are the cornerstone of treatment for this condition, often requiring extended use. A concerning issue is the decline in the efficacy of commonly used antimicrobials like enrofloxacin, likely due to increasing pathogen resistance. Authors have highlighted the effectiveness of amikacin in cases where other drugs have failed [3,5], and this drug has shown success in treating quinolone-unresponsive cases [8]. All *Escherichia coli* isolated from intestinal biopsy samples in dogs with ulcerative colitis were sensitive to amikacin [19], however, potential nephrotoxicity associated with amikacin must be considered. Fortunately, no signs of nephrotoxicity were observed in this case.

Hydrolyzed and fiber-enriched diets can serve as adjuvant therapy with a range of benefits including high digestibility, increased energy content, prebiotic effects, reduced allergenic stimulation in the gastrointestinal tract, and improved motility. There is currently no scientific evidence supporting the superiority of homemade (natural) diets over commercially manufactured diets (kibble). Therefore, the choice should be guided by the owner’s preferences and the patient’s specific needs [4,8]. Nonetheless, the incomplete response to a hydrolyzed diet and standard treatments including mesalazine, probiotics, and fiber was observed in the case reported is a common feature in animals diagnosed with granulomatous colitis [5,7].

According to the literature [24,25], the dysbiosis index is considered the gold standard for assessing eubiosis or dysbiosis, as only clinical improvement or serum folate increase are currently considered inconsistent indicators. The fecal transplant utilized in this study adhered to recently established methodologies regarding donor criteria and application guidelines [21]. The increased expression of *Clostridium hiranonis* played a decisive role in the patient described in this report. This bacterium is considered beneficial and inhibitory toward other bacteria deemed pathogenic and toxigenic [25,26].

The role of fecal microbiota transplantation in managing recurrent, multidrug-resistant granulomatous colitis in French Bulldogs has not been previously documented. However, its application has been explored in cases of parvovirus [27] (PEREIRA et al., 2018), clostridiosis [28], and as an adjuvant therapy against chronic enteropathy responsive to immunosuppressants [18,21]. The value of FMT lies in its potential to reduce antibiotic use, which can induce dysbiosis, diminish intestinal immunity, pose toxicity risks, and necessitate costly monitoring [16].

## 4. Conclusions

Fecal microbiota transplantation was a viable and effective approach able to resolve the clinical signs of granulomatous colitis in a French Bulldog. This innovative method aligns with the current scientific understanding that chronic gut diseases are often associated with significant dysbiosis, which cannot always be fully addressed with standard antibiotic therapies. Moreover, the condition was partially responsive to antibiotics, but FMT successfully reduced the need for their use. As a promising alternative, fecal microbiota transplantation offers the potential for more consistent and favorable outcomes and could become a practical, routine therapy in clinical practice as well as a viable option for recurrent treatment.

## Figures and Tables

**Figure 1 microorganisms-13-00366-f001:**
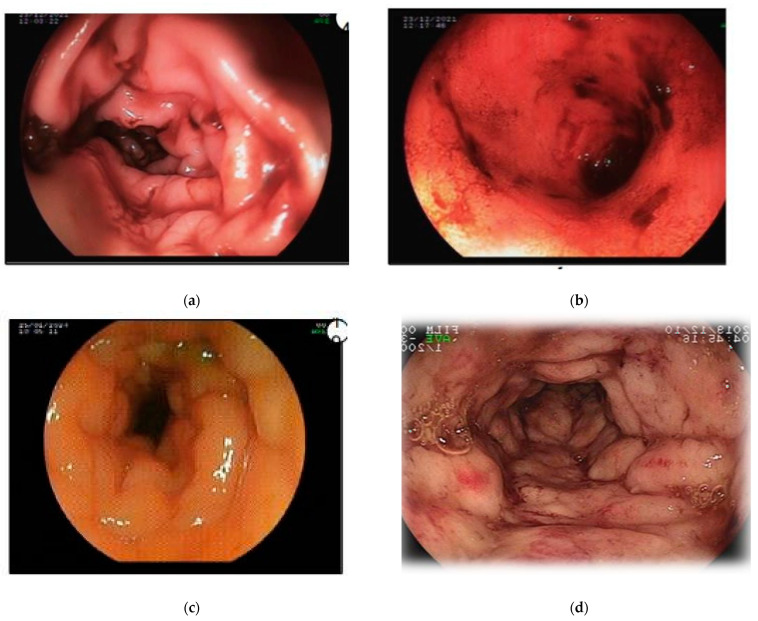
Colonoscopy images showing wall edema, bleeding, lumen stenosis, and hyperemia in the (**a**) cecum, (**b**) ileum, (**c**) colon, and (**d**) rectum. Due to significant colonic wall edema, two different types of equipment were used (sizes and calibers/thicknesses) to obtain adequate images.

**Figure 2 microorganisms-13-00366-f002:**
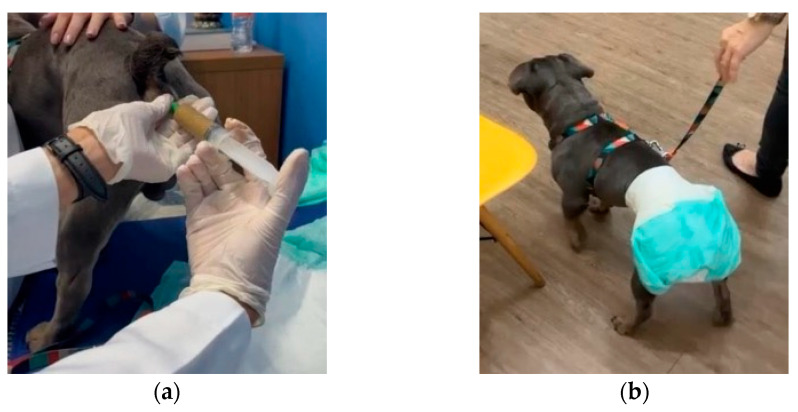
Fecal transplantation: (**a**) patient during fecal infusion via rectal probe and (**b**) after the infusion.

**Figure 3 microorganisms-13-00366-f003:**
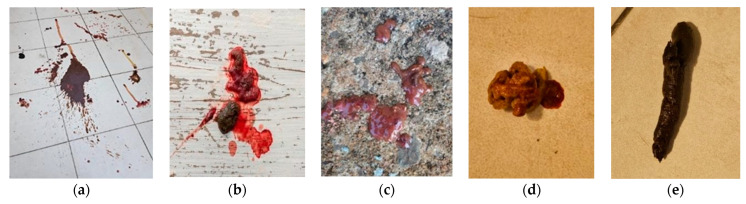
Fecal score before fecal transplantation: (**a**) watery diarrhea, (**b**) hematochezia, (**c**) mucus and blood in feces, and 48 h after fecal transplantation (**d**) with format and no mucus, but a small amount of blood, and (**e**) normal feces without blood and mucus.

**Table 1 microorganisms-13-00366-t001:** Patient’s results (result in log DNA) for the dysbiosis index before and after fecal microbiota transplantation (FMT).

	Before FMT	After FMT	Reference
*Faecalibacterium*	4.7	5.3	3.4–8.0
*Turicibacter*	5.3	4.3	4.6–8.1
*Blautia*	10.6	10.4	9.5–11.0
*Fusobacterium*	8.7	9.1	7.0–10.3
*Clostridium hiranonis*	0.1	6.3	5.1–7.1
*Streptococcus*	4.3	4.8	1.9–8.0
*Escherichia coli*	7.0	5.5	0.9–8.0
Dysbiosis index	4.1	−2	<0 normal 0–2 mildly increased >2 significantly increased

Data provided by the Gastrointestinal Laboratory, Texas A&M University, College Station, TX, USA [20].

**Table 2 microorganisms-13-00366-t002:** Patient’s results for the dysbiosis index after fecal microbiota transplantation.

	Result (log DNA)	Reference
*Faecalibacterium*	5.3	3.4–8.0
*Turicibacter*	4.3	4.6–8.1
*Blautia*	10.4	9.5–11.0
*Fusobacterium*	9.1	7.0–10.3
*Clostridium hiranonis*	6.3	5.1–7.1
*Streptococcus*	4.8	1.9–8.0
*Escherichia coli*	5.5	0.9–8.0
Dysbiosis index	−2	<0 normal 0–2 mildly increased >2 significantly increased

Data provided by the Gastrointestinal Laboratory, Texas A&M University, College Station, TX, USA [20].

## Data Availability

The original contributions presented in this study are included in the article. Further inquiries can be directed to the corresponding authors.

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
