# Peer review of "Fecal Microbiota Transplantation as a Treatment for Granulomatous Colitis in a French Bulldog: A Case Report"

_microorganisms, 2025, doi:10.3390/microorganisms13020366_

Round 1

Reviewer 1 Report

Comments and Suggestions for Authors

The results of treatment of a French bulldog with ulcerative colitis, dysbiosis and resistance to conventional treatment are presented in the article "Fecal microbiota transplantation as a treatment for granulomatous colitis in a french bulldog: a case report". The dog is sensitive and partially responsive to amikacin. After fecal transplantation, the patient's index steadily improved and was under control. This method improved clinical results and restored the microbiota without requiring further antibacterial treatment. Without a doubt, practice and basic science will benefit from the results of this study. After a few minor revisions, the article can be published.

Overall, the article is well written. As a minor correction, I would recommend rewriting the conclusions. This section should not just be a summary; it should contain a take-home message.

Reviewer 2 Report

Comments and Suggestions for Authors

Romano et al. evaluated the effects of fecal microbiota transplantation on granulomatous colitis using a french bullgod animal model and concluded this method can be used for microbiome restoration. This is a ood study on an important topic. However, I have some suggestions, in my opinion, that may improve this manuscript.

1. Please demonstrate the microbiome composition before and after the fecal microbiota transplantation in the animal model.

2. How to make sure the fecal microboota transplantation worked in this study?

3. The proper control for fecal microbiota transplantation is missing. 

Reviewer 3 Report

Comments and Suggestions for Authors

Fecal microbiota transplantation as a treatment for granulomatous colitis in a French bulldog: a case report

The case report of a French Bulldog diagnosed with ulcerative colitis accompanied by dysbiosis and refractory to standard treatments but sensitive and partially responsive to amikacin. The patient achieved control and sustained improvement in fecal scoring following fecal transplantation. The authors suggested that this approach prevented additional antibiotic therapy, ensuring clinical amelioration alongside microbiome restoration.

L22, 29, 54: be consistent with the bacteria name E. coli

L43-45: be specific about the species, is this in dogs?

The introduction has useful information, refining is suggested.

L113: The patient’s dysbiosis index was assessed for the current case. Clarify

A brief description of the methods used is required.

L122: write the reference

L125: more information is required about the donor such as breed, age, weight, etc.

L150-151: revise

L165: According to Chaitman & Gashen [24] and Pilla and Suchodolski [25], be consistent with references within the text.

L175: (PEREIRA et al., 2018), be consistent

L181-182: revise conclusion, don't generalize

Round 2

Reviewer 2 Report

Comments and Suggestions for Authors

No further comments